# p53 Modulates Radiosensitivity in Head and Neck Cancers—From Classic to Future Horizons

**DOI:** 10.3390/diagnostics12123052

**Published:** 2022-12-05

**Authors:** Camil Ciprian Mireștean, Roxana Irina Iancu, Dragoș Petru Teodor Iancu

**Affiliations:** 1Department of Oncology and Radiotherapy, University of Medicine and Pharmacy Craiova, 200349 Craiova, Romania; 2Department of Surgery, Railways Clinical Hospital Iasi, 700506 Iași, Romania; 3Oral Pathology Department, Faculty of Dental Medicine, “Gr. T. Popa” University of Medicine and Pharmacy, 700115 Iași, Romania; 4Department of Clinical Laboratory, “St. Spiridon” Emergency Universitary Hospital, 700111 Iași, Romania; 5Oncology and Radiotherapy Department, Faculty of Medicine, “Gr. T. Popa” University of Medicine and Pharmacy, 700115 Iași, Romania; 6Department of Radiation Oncology, Regional Institute of Oncology, 700483 Iași, Romania

**Keywords:** p53, radiosensitivity, radioresistance, radiotherapy, head and neck cancers, HNSCC

## Abstract

p53, initially considered a tumor suppressor, has been the subject of research related to cancer treatment resistance in the last 30 years. The unfavorable response to multimodal therapy and the higher recurrence rate, despite an aggressive approach, make HNSCC a research topic of interest for improving therapeutic outcomes, even if it is only the sixth most common malignancy worldwide. New advances in molecular biology and genetics include the involvement of miRNA in the control of the p53 pathway, the understanding of mechanisms such as gain/loss of function, and the development of different methods to restore p53 function, especially for HPV-negative cases. The different ratio between mutant p53 status in the primary tumor and distant metastasis originating HNSCC may serve to select the best therapeutic target for activating an abscopal effect by radiotherapy as a “booster” of the immune system. P53 may also be a key player in choosing radiotherapy fractionation regimens. Targeting any pathway involving p53, including tumor metabolism, in particular the Warburg effect, could modulate the radiosensitivity and chemo-sensitivity of head and neck cancers.

## 1. Introduction:

Besides a certain prognostic improvement in the last decades, the loco-regional failure rate remains significant in head and neck cancers, justifying an increased interest in research on this topic, both for the identification of new biomarkers and for therapeutic targets but also to identify methods in order to reduce the treatment associated adverse events. Although it is the sixth leading cause of malignancy worldwide, head and neck squamous cell carcinoma (HNSCC) is notable for the higher rates of therapeutic failure, especially due to loco-regional recurrence. Human cancers are associated with the inactivation of one or more components of the p53 pathway, but HNSCC is notable for a high rate of p53 pathway inactivation caused by TP53 gene mutation. Among the exceptions are head and neck cancers associated with human papillomavirus (HPV), a subtype of HNSCC associated with virus-induced inactivation of the p53 pathway [1].

Discovered in 1979, p53 is a protein of about 53 kDa expressed highly in cancer cells. p53 is considered today not only to be involved in DNA damage but is also a mediator of responses to cellular stress and is associated with sensitivity to irradiation and chemotherapy of malignant tumors. Scientific knowledge of current molecular biology has made a decisive contribution to understanding the mechanisms involved in the p53 pathway by identifying an increasing number of post-transcriptional targets and also by understanding p53-mediated apoptotic mechanisms. Testing and validation of agents and mechanisms targeting mutant and wild-type variants of p53 open new perspectives for improving the therapeutic ratio of oncological therapies by enhancing tumor destruction and simultaneously protecting healthy tissues. Recently, immune checkpoint inhibitors (ICIs) entered the therapeutic spectrum of HNSCC and the potential biomarker value of the TP53 gene mutation, the most common genetic mutation in these cancers, associated with the accumulation of p53 in the malignant cell, is investigated in correlation with tumor mutation burden (TMB) and tumor neo-antigens (TNA), both for HNSCC primary tumor and for distant metastases. Three therapeutic strategies are proposed for the restoration of p53 function and consequently for the improvement of therapeutic results in HNSCC: targeting the degradation or direct inhibition of wild-type p53 (WT), reactivation of transcriptional activity by binding mutant p53 and restoration of WT p53 status [2,3,4].

A search in the PubMed^®^ database was performed using the terms “p53”, “head and neck cancer,” or HNSCC and radiosensitivity/radioresistance or chemoresistance. Subsequently, searches were attempted with each of the HNSCC types according to the anatomical structure it affects, and the studies considered relevant were included in a narrative review.

## 2. p53 and Therapeutic Interactions in Cancer—A Brief Description

The main mechanism of p53 action is tumor suppression by transcriptional regulatory apoptosis, with loss of p53 function being detected in approximately 50% of cancers, especially in solid tumors. p53 loss of function is also considered an early event of carcinogenesis. In response to DNA damage, accumulation of p53 can occur in the cell nucleus, inducing the mechanisms of apoptosis and limiting cell division. As a consequence of these processes, DNA-induced lesions propagation is stopped. Mutant p53 provides glucose and nutrient resources to prevent reactive oxygen species (ROS) -mediated cell destruction. In the case of WT p53, the effect is the opposite, targeting metabolism being a topic of interest to destroy the malignant cell by depriving resources of tumor cell energy. Even if the main interaction with DNA is coordinated, especially by the central portion that mediates the DNA binding process, the N-terminal and C-terminal ends are involved in the transactivation capacity, respectively, and the post-translational changes of p53 [5,6,7]. 75% of p53 mutations are represented by missense mutation, followed by frame-shit insertion/deletion, nonsense mutation and silent mutation in proportions of 9%, 7% and 5%, respectively [8]. Both ionizing radiation and DNA chain-inducing chemotherapy agents can activate p53-mediated cellular mechanisms via mutated ataxia gene (ATM), a p21 mediates cell cycle arrest between the G1 and S phases, or via transcriptional induction of PUMA and oligomerization of Bax, in this case, the radiobiological effect is linked to the modulation of apoptosis. Considered the “guardian of the genome” for its blocking feature in cells with DNA strands affected by progression at the G1/S transition or even by inducing apoptosis, p53 with compromised function is a true modulator of chemotherapy and radiotherapy sensitivity, with evidence of both augmentation and the reduced amplitude of responses to physical and chemical agents [4,9,10].

The mutant variant has the opposite effect to wild type (WT) p53 by the “gain of function” (GOF) phenomenon, thus contributing to the increase of the resistance to treatment of the malignant cell but also promoting the tumor progression. Recent data highlight that the GOF phenomenon has not only the effect of losing cell homeostasis and inactivating the mechanisms that suppress tumors, but the p53 amount proves to be a true “hallmark of cancer,” promoting tumor progression and invasion. About 28% of the mutant p53 are associated with eight gene mutations, most commonly situated on six codons, the positions R175H and R273 being considered “hot spots codons positions.” R175H and R273 are associated with the loss by p53 of the DNA binding interface. The p53 mutant is also involved in acquiring by malignant cells with mesenchymal characteristics, detaching from the substrate and migrating. The ability of the mutant p53 to induce epithelial-mesenchymal transition (EMT) has been demonstrated for 10 years in prostate cancer cells by up-regulation of TWIST1, one of the transcription factors involved in EMT. The GOF effect of mutant p53 implies another “hallmark of cancer,” inflammation, demonstrating the ability of mutant p53 to alter the reactivity of the tumor microenvironment, promoting chronic inflammation, especially by modulating the TNF-alpha-activated NF-κB regulator [11,12,13].

Another pathophysiological phenomenon in which mutant p53 is involved in the modulation of tumor metabolism. Physiologically, malignant cells use glycolysis more than the oxidative pathway, a metabolic shift known as the Warburg effect. In contrast to p53, a tumor subtype associated with the Warburg inhibition phenomenon, mutant p53 intensifies glycolysis involving the GLUT1 transporter, mediated by RHOA/ROCK signaling. Protease-induced tumor microenvironment acidification also promotes tumor proliferation and chemo-resistance. In summary, p53, known as a tumor suppressor gene, becomes an oncogene, a mutant variant that, by a gain of function (GOF), alters the secretion of enzymes implicated in the modulation of extracellular matrix components mediates crosstalk between tumor cells and the microenvironment, acidifies the tumor microenvironment, and alters inflammatory cytokine secretion [13,14].

More than 20 years have passed since there was scientific evidence involving a mutation in the p53 gene correlated with cancer resistance to chemotherapeutic agents and radiation therapy. Today, micro-RNAs (miRNAs), a class of small non-coding RNA molecules, are known to be implicated in regulating the expression of genes that involve division, differentiation, metabolism, and malignancy, and p53 is a key player in these interactions as well [15,16].

## 3. p53 and Chemotherapy—An Old but Always Up-to-Date Collaboration

A hypothesis issued by Chresta et al. mentions the inability of cytotoxic chemotherapy to destroy metastatic malignant cells in relation to the inability of these cells to engage in apoptosis after drug-induced lesions. Noting that 60–70% of bladder carcinomas have p53 amounts, the authors remark that cells containing a p53 WT pass through a G1 checkpoint after exposure to topoisomerase II (etposide) inhibitors, cell cycle phase transition that allows them to repair DNA-induced lesions. Cell lines containing mutant or dysfunctional p53 were associated with high expression of bcl2 and bcl-XL apoptosis suppressors and low bax levels, a factor favoring apoptosis. This set of gene expressions proves the involvement of p53 in resistance to chemotherapy. The study by Burger et al. concludes that hypersensitivity to Cisplatin does not need to be correlated with p53 WT status, thus demonstrating that platinum sensitivity in testicular germ cell tumor cells is not associated with Bcl-2 and Bax expression. Loss of p53 function is correlated with increased cisplatin cytotoxicity in ovarian cancer cell lines, associated with decreased cisplatin DNA adduct repair and loss of G(1)/S control checkpoint [17,18,19].

For Carboplatin-treated cervical cancer, p53 is an upstream regulator of extracellular recognition kinase (ERK) activation, thus promoting the induction of apoptosis. p53 is a critical player in cancer stem cell (CSC) activation in colorectal cancer treated with 5-fluorouracil (5-FU), a process mediated by the WNT/β-catenin pathway. The study by Cho et al. proposes a WNT inhibitor in combination with 5-FU to overcome treatment resistance involving the p53 pathway [20].

TP53 inactivation is associated with taxane resistance in ovarian cancer cells, and hyper-activation via the WNT/β-catenin pathway may also be used in ovarian cancer as a therapeutic strategy. The P53 mutation is associated with lower survival rates in bile duct cancers but also with resistance to Gemcitabine-based chemotherapy. Indirect targeting of checkpoints Chk1, ATR and Wee1 are proposed therapeutic strategies for mutant p53 tumors. In the case of p53 WT tumors, inhibition of its negative regulators MDM2 and WIP1 is proposed as a future therapeutic strategy [21,22,23].

The p53 mutant may cause chemo-resistance in colon cancers due to the inactivation of PUMA transcription. Hunag and collaborators consider the opportunity to identify therapeutic options in the case of mutant p53 cancers in order to overcome the mechanisms of resistance to chemotherapy [24].

Tumor suppressor p53 is frequently inactivated by cancer cells, the mechanism often being that of missense mutations. In 30% of cases, the transcriptional activity is completely lost, and consequently, resistance to chemotherapy secondary to p53 mutations is generated. 70% of non-hotspot mutants are associated with p53 partial loss of function (LOF), and the consequence is a residual transcriptional activity. Preclinical models have shown that partial LOF is associated with maintaining a rate of response to chemotherapy and the mechanisms of cell death by apoptosis. Klimovich et al. demonstrate that p53 non-hot spot mutations associated with partial LOF should be distinguished from other mutations, not associated with chemo-resistance. Thus, this type of mutation has a positive impact on the survival of patients treated with chemotherapy [25].

More than two decades ago, Mueller et al. mentioned the dual role of wt p53 as a trigger for apoptosis and an initiator of DNA damage repair. The authors also hypothesized that chemotherapy, especially Cisplatin and other agents which act at the DNA level, may have an increased therapeutic benefit in cases of p53 mutant tumors [26].

Cytotoxic agents such as platinum salts, alkylating agents, anthracyclines, antimetabolites and biological therapies such as anti-estrogens and EGFR inhibitors are mentioned among agents whose clinical response is modulated by p53 status. Small molecules such as PRIMA-1, MIRA-1, thiosemicarbazone family derivatives and p53-MDM2 axis controlling compounds are proposed for possible strategies in order to improve the response to chemotherapy and biologic target therapy [27].

Doxorubicin resistance appears to have a different mechanism compared to the p53-mediated response to other alkylating agents. p53 deficient fibroblast mice are more resistant to Doxorubicin treatment than wt p53 cells. These data are in contrast to the results obtained by exposing p53-deficient cells to chemotherapy, considered more resistant to genotoxic agents. Dunkern et al. hypothesized that Doxorubicin resistance is associated with reduced production of DNA strand breaks, inhibition of apoptosis not being the main mechanism involved. Simultaneously, analyzing the role of p53 in Gemcitabine-mediated cytotoxicity and in the radiosensitization of RKO cells of colon cancer, the study demonstrated that p53 could be a mediator of Gemcitabine sensitivity, but the effect is considered minor. Surprisingly, the combination of radiation therapy with Gemcitabine has been proven to be detrimental in the study of Salem and collaborators for mutant p53 breast cancer cells. Jackson et al. reported the induction of senescence and the prevention of mitotic catastrophe as the basis of the differentiated response of breast cancer cells to Doxorubicin, depending on the status of p53. CHEK2 mutations affecting kinase activity appear to have a synergistic effect with the p53 mutation in modulating Epirubicin sensitivity in breast cancer [28,29,30,31,32,33]

The absence of functional p53 has been associated with increased sensitivity to Temozolomide in glioblastoma cells, with p53 status being a predictive factor independent of MGMT mutation. p53 has not been identified as an independent prognostic factor in ovarian cancer, but p53 impairment has been associated with platinum resistance and lower survival rates. Extracellular recognition kinase (ERK) modulates p53 cascades and is involved in the apoptosis of cervical cancer cells exposed to Carboplatin treatment. The negative effect of the p53 mutation was not overcome by the addition of Paclitaxel to Carboplatin in ovarian cancer [20,34,35,36].

Rab-coupling protein (CPR) is known to play a role in endosomal recycling and in integrin and receptor tyrosine kinase signaling. The interaction of P-glycoprotein (P-gp) with CPR is associated with resistance to Cisplatin and Etoposide. Mutant, but not null p53 expression enhances the co-localization of RCP and P-gp and thus promotes the mechanism of delivery of P-gp in plasma membranes, being a radioresistance factor [36]. Souza et al. proposed a modern perspective regarding the mechanisms of p53-mediated chemo-resistance, the one based on the crosstalk between tumor and tumor microenvironment. Thus, in the modern view, the p53-mediated chemo-resistance effect is a multicellular/tissue-level phenomenon. Cross-mediated p53 targeting between the cell and the tumor microenvironment may therefore be a promising strategy to counteract cancer resistance to chemotherapy [37].

## 4. p53—Orchestrator of Cancer Radiosensitivity

Ionizing radiation induces double-stranded breaks (DSB) DNA and consequently will result in the activation of checkpoints, initiating signals that lead to malignant cell death or survival. The ability to modulate the repair of cell lesions caused by p53 irradiation is already recognized as a decisive factor in tumor radiosensitivity [38].

The control of G1 and G2 checkpoints, and especially G1 arrest, is considered associated with p53 status. If malignant cells express p53 normally, there is no evidence of G1/S cell cycle arrest, but phosphorylated p53 has the ability to modulate p21-mediated G1/S arrest. The down-regulation of mitochondrial transcription factors is associated with radiosensitization, with p53 being involved in the signaling pathway like ATM/p53/p21 and ATM/CHK2/CDC25C. Note that one of the two control pathways of G1/S cell checkpoints is mediated by ATM phosphorylated p53 [39,40,41]. Hinata and collaborators demonstrate on bladder cancer cell lines that ionizing radiation-induced p53-mediated apoptosis in WT bladder cancer cells but not in p53 mutant cancer cells [40,42].

Li-Fraumeni syndrome is associated with a p53 germ-line mutation, characterized by an increased susceptibility to cancer, but also with a higher rate of severe toxicity. However, Wong et al. report the lack of late side effects and a second radio-induced cancer for a heavily treated patient with three radiotherapy sequences for various cancers. The authors note the possible protective effect of the p53 mutation for irradiation-related toxicities [43].

The p53 regulatory agents have been shown to be effective in the protection against acute irradiation syndrome when high-LET heavy ion radiation (> 85 keV/μm) has been used [44].

Gastrointestinal toxicity, one of the factors limiting the dose of irradiation in lower abdominal tumors, is a goal to increase the therapeutic ratio of these tumors. The study of Pant et al. demonstrated the increase in p21 activity induced by p53 as being associated with a reduction in the adverse effects of irradiation. Consequently, the authors proposed a pharmacological agent that interrupts the p53-Mdm2 interaction for the reduction of normal tissue radiosensitivity levels. Cell cycle, proliferation and DNA repair are modulated by irradiation via p53, EGFR and ERCC1 expression in human cervical cancers but not in cervical cancer cell lines. The study results demonstrated the involvement of p53 in the radioresistance of cervical cancer [45,46].

In the context of precision radiotherapy materialized by choosing the optimal fractionation, Anbalagan et al. propose the concept of different susceptibility to the same fractionation scheme modulated by intact non-homologous end-joining (NHEJ) and by wild-type p53 WT. The study results highlight split-dose recovery associated with WT p53, but the loss of the sparing effect of a smaller dose per fraction is associated with Li-Fraumeni fibroblasts due to a defective G1/S checkpoint and a large S/G2 component. NHEJ-deficient cells have no split-dose recovery effect, and p53-defective cells are considered more sensitive to irradiation [47].

Restoration of hypoxia-induced apoptosis in mutant p53 tumors is currently considered a therapeutic strategy. The study proposed by Leszczynska et al. mentions hypoxia-inducible pro-apoptotic factors and an unfavorable prognosis associated with their deregulation, but also a possible strategy to increase tumor radiosensitivity in hypoxic regions with p53 deficiency by pharmacological inhibition of AKT [48].

The dynamics of p53 during irradiation may be a factor that could influence the DNA lesion repair capacity via feedback mechanisms linked to p53 target genes, implying cell death and radioresistance. The response to irradiation itself modulates the behavior of p53, radiosensitive and radioresistant tissues being associated with prolonged p53 signaling, respectively, with transient p53 activation. Alteration of p53 dynamics is a strategy to promote radiosensitivity proposed by Stewart-Ornstein et al. [49].

p53 demonstrates the ability to modulate radiosensitivity by two distinct mechanisms, demonstrated with a preclinical model of lung cancer cells. Inhibition of autophagy, a pro-survival mechanism, simultaneously with the promotion of apoptosis, is a mechanism that promotes cell death. MDM2 and P21 higher expression are associated with p53-mediated radiosensitivity. Radiation-induced autophagy has also been increased in p53-expressing cells, demonstrating the involvement of two distinct substrates in the radiobiology of lung cancer. Not only the loss of p53 function but also the loss of p73 function has been associated with reduced chemo-sensitivity and radiosensitivity. Taking into account these data, Cuddihy and collaborators proposed the use of pre-therapeutic “molecular therapeutic ratio” as a strategy for treatment personalization based on radiosensitivity and chemo-sensitivity. Analyzing the clinical phenotype of TP53 mutations in radiotherapy patients for rhabdomyosarcoma and Ewing’s sarcoma in a patients lot including 397 cases, most Ewing’s sarcomas, the study by Casey et al. demonstrated a reduced tumor control in the case of any types of sarcomas for both TP53 mutant cases and p53 pathway alteration. Multivariate analysis also identified gross tumor, histology, biological dose, and radiotherapy intent, along with p53 status as predictors of radioresistant phenotype. Non-endometrioid endometrial carcinoma has been associated with increased radiation resistance in the case of abnormal accumulation of p53. Thus, the cases with overexpression of p53 had an unfavorable prognosis and an increased radioresistance profile if adjuvant radiotherapy was administered. No differences were observed between cases with different expressions of p53 for adjuvant chemotherapy. The study demonstrates the involvement of p53 overexpression in radioresistance but not in the chemo-resistance of non-endometrioid endometrial carcinoma [50,51,52,53].

p53 downstream is considered pivotal in RIG-I, a cytosolic immune receptor ligand associated with radiotherapy in malignant melanoma multimodal treatment in order to overcome radioresistance. A novel radiation-induced mutation at the end of the DNA binding domain of p53 has not been demonstrated to play a role in radiation-induced radioresistance on lung cancer cell lines, as evidenced by Sun and collaborators [54,55].

Both p53 signaling and tumor protein p53 binding protein 1 (TP53BP1), along with non-homologous end-joining (NHEJ), are involved in various mechanisms of non-small cell lung cancer (NSCLC) response of multi-fractions irradiation, so as demonstrated in preclinical models. These data suggest the use of both p53 and TP53BP1 expression as radiosensitivity biomarkers with a possible future role in the clinical therapeutic decision [56,57].

Inhibition of the interaction between MDM2 and X-p53 is a strategy with the potential to be introduced into the routine of clinical practice. MDM2 inhibitors are considered the potential for radiosensitization of glioblastoma multiform (GBM) cells by restoring p53 functions and converting mutant p53 to a wild-type variant. Removal of senescent cells by macrophages is another mechanism considered with potential for radiosensitizing GBM involving p53. Targeting of senescence-associated p53 isoform by reducing the expression of D133p53a and consequently by reducing cell rescue from senescence has the end result of reducing a decreased ability of cells to repair DNA damage and consequently a superior radiosensitivity [58,59].

## 5. p53, Chemo-Sensitivity, Radiosensitivity and Tumor Metabolism—An Alliance Finally Revealed in HNSCC

In 1924, Otto Warburg first mentioned the metabolic peculiarity of cancer using glycolysis to generate adenosine triphosphate, nucleotides, lipids, and amino acids to ensure the increased energy and nutritional necessity associated with rapid tumor proliferation. Glycolysis is not only a key player in providing an energy substrate, it involves the activation of oncogenes such as phosphatidylinositol 3-kinase (PI3K) but also the modulation of HIF-1A and modifies the tumor microenvironment (TME). Thus, tumor metabolism may be indirectly correlated with the mechanisms of tumor response to chemotherapy [60,61,62].

The tumor cell interacts with the TME for adaptive purposes, resistance to therapies being the result of such a phenomenon. The Warburg effect, known for almost 100 years, modifies the cellular metabolism of aerobic glycolysis, having the role of generating vital elements for the survival of the tumor cell. Not only is the Warburg effect modulated by p53, but also the metabolism of lipids and amino acids, the generation of growth factors and reactive oxygen species are involved in this process. By regulating a large number of genes (more than 500) capable of manipulating tumor metabolism, the wild-type or mutant status of p53 may have different effects on tumor metabolism and, consequently, indirectly on resistance to therapeutic agents. Manipulation of tumor metabolism could be done either by drugs that interact with metabolic processes or by inducing p53 expression that generates senescence, autophagy and apoptosis via the PI3K/AKT/mTOR and ROS pathways. The mutant P53 stimulates the Warburg effect by promoting GLUT1 translocation to the plasma membrane. One of the strategies to limit mutant p53 GOF to promote the Warburg effect is to inhibit RhoA/ROCK/GLUT1 signaling. Also, targeting tumor cell glycolysis has the effect of limiting p53 mutant-mediated tumor genesis. The succinate dehydrogenase 5 (SDH5) protein, also named succinate-coenzyme Q reductase (SQR), an enzyme complex with the catalytic role of succinate oxidation into fumarate in the Krebs cycle, inhibits p53 degradation [62,63,64,65,66].

HNSCC expressing the TP53 mutation is associated with a higher level of radioresistance compared to HNSCC wild type TP53, as demonstrated by Sandulache et al., the sensitivity to glycolytic inhibition being explained by the decrease in mitochondrial complex II and IV activity in cases expressing the TP53 mutation. HNSCC cases associated with wild-type P53 can be sensitized to glycolytic inhibition using Metformin breathing inhibition in order to increase radiosensitivity. Loss of p53 function causes the Warburg effect in HNSCC, and this metabolic vulnerability can be exploited to increase radiosensitivity. Being more dependent on glycolysis, without having the ability to use oxidative phosphorylation, these cases can benefit from treatment with glycolytic inhibitors. Wilkie and collaborators even proposed a stratification of patients based on these concepts. 2-deoxyglucose, a glycolytic inhibitor and N-acetylcysteine (NAC), an anti-oxidant, were evaluated in vitro in association with radiotherapy in head and neck cancers, evaluated by flow cytometry and apoptosis and reactive oxygen species (ROS). Only in p53 mutant cases was the response to radiotherapy correlated with glycolytic inhibition, with the addition of NAC having a reverse effect [67,68,69].

In the case of cancers associated with HPV infection, the E6 and E7 oncoproteins inactivate p53 and pRb and, by activating the mammalian target of rapamycin (mTOR) pathway, activate the Warburg effect, LDHA accumulation and lactate production. Thus, by inactivating p53, E6 and E7 indirectly affect protein synthesis mechanisms via the PPP pathway and HIF-1α, influencing angiogenesis, tumor progression and glycolysis through this pathway. The higher levels of glycolysis in HPV-negative head and neck cancer cells in relation to HPV-associated HNSCC justify the testing of glycolytic inhibitors, especially for negative HNSCC. According to Sallter et al., pre-treatment acidification of the tumor microenvironment determined by lactate and lactate pyruvate is associated with radioresistance. Thus the Warburg effect could be correlated with radioresistance. TP53 mutation, HIF-1, TKTL, GLUT-1, LDH-A, HKII and MCTs were identified as modulating factors of the Warburg effect and of radioresistance and chemoresistance in HNSCC [70,71,72,73,74,75,76,77,78].

Cisplatin, one of the cornerstones of HNSCC systemic treatment, is involved in protein synthesis modulated by PPP pathways, glycolysis and the Krebs cycle. Yu and colleagues consider that targeting metabolic pathways is a future strategy for reducing chemoresistance beyond the simple inhibition of glycolysis. In head and neck cancers, even chronic exposure to Cisplatin could induce a metabolic reprogramming of the tumor cell, neutralizing oxidative stress [79].

## 6. MicroRNAs—New Kids on the HNSCC Block—Focus on p53 Mediated Radiosensitivity and Chemo-Sensitivity

MiRNAs, a class of small non-coding RNA molecules, are indirectly involved in modulating radiosensitivity by regulating processes such as cell division, differentiation, tumor metabolism, apoptosis and gene expression. The interaction of p53 miRNAs is essential in fine-tuning tumor responses to irradiation. miRNAs bind to the 3′UTR of p53 mRNA and could down-regulate genes such as p21 or MDM2 and consequently generate radioresistance by inducing arrest in the G1 phase. MicroRNA (miRNA, miRs) controls radiobiological processes involved in tumor sensitivity to DNA radiation damage repair, modulation of apoptosis, control of cell cycle checkpoint, and signal transduction pathways tumor microenvironment. MiR-300 expression was analyzed in radiation-treated lung cancer cells, and the ability of miR-300 to regulate DNA damage, cell cycle arrest, apoptosis, and cellular senescence associated with irradiation was also assessed. Up-regulation of miR-300 induced by irradiation has been associated with improved DNA damage repair, but also with cell cycle arrest in G2 and inhibition of apoptosis. p53 and apaf1 were also targets of miR-300, so the mechanisms for regulating the radiosensitivity of lung cancer of this miRNA are, therefore, complex. By activating the PBK-dependent p53 pathway, MicroRNA-372 has a radiosensitizing role in nasopharyngeal cancer but is also an inhibitor of cell invasion and metastasis. PI3-K/Akt, NF-κB, MAPK, and TGFβ are miRNA-modulated pathways involving radiosensitivity, as mentioned by Zhao et al., and p53 is one of the key molecules of these mechanisms. P53-modulated chemo-sensitivity also involves p63 and p73-bound transcriptional agents, which have different functions in the survival and progression of squamous cell carcinomas. miR-193 a is involved in p63 suppression and p73 activation, the mechanisms being independent of p53 status. These data argue for the use of this miRNA as a biomarker for strategies to modulate sensitivity to chemotherapy in p53-deficient tumors. By activating the PBK-dependent p53 pathway, MicroRNA-372 has a radiosensitizing role in nasopharyngeal cancer but also is an inhibitor of cell invasion and metastasis. p53/miR-149-3p/PDK2 axis regulates colorectal cancer chemo-sensitivity via glucose metabolism. Twelve miRNAs are correlated with p53, and two miRNAs are associated with irradiation or both with irradiation and p53 in an analysis that included human colon carcinoma cell lines [4,80,81,82,83,84,85,86].

Cancer stem cells (CSCs) are involved in mechanisms of radioresistance and chemoresistance as well as progression, metastasis, and evasion of the immune system in HNSCC, being correlated with an unfavorable prognosis and recurrence of the disease. miRNAs have been associated with the acquisition of stem-like properties of tumor cells in HNSCC, the acquiring of plasticity and resistance to stress being some of these characteristics. miR-125a suppresses the expression of p53 and thus maintains these properties of the cancer cell. Inhibition of PDZ-binding kinase (PBK) and p53 activation has the effect of radiosensitization of nasopharyngeal tumors, a process regulated by miR-372. miR-200 is involved in reducing EMT mediated by p53 [87,88,89].

The let-7c miRNA family was identified as being associated with stemless, radioresistance and chemo-resistance in oral cancer. Peng et al. propose targeting this class of miRNAs for the prevention of recurrence in oral cavity squamous cell carcinoma. miR-17-5p expression decreases with betel nut chewing, a well-known risk factor for oral cancer. An OC3 xenograft tumor model, but also an in vivo model, was used to evaluate the role of p53 in irradiated cells. p53 expression was associated with G2/M arrest induced by irradiation. miR-17-5p can both inhibit and potentiate proteins related to apoptosis, and the regulation that this mi-RNA exerts on p53 modifies radiosensitivity [90,91].

## 7. p53 in Immunotherapy ERA

p53 expression is associated with an increased load of neoantigens, chemokines, proinflammatory mediators and an increased tumor burden, as demonstrated by Lin and collaborators. By correlating p53 with TME characteristics, p53 is also associated with the response to ICIs therapy, but the authors also propose the use of p53 status as a biomarker of treatment response. In HNSCC metastatic and recurrent settings KEYNOTE-048 trial imposed the combination of Pembrolizumab plus chemotherapy as superior to the EXTREME regimen (cis- or carboplatin, 5-fluorouracil (5-FU) and cetuximab). Even if the Javelin Head Neck 100 trial failed to demonstrate a PFS and OS benefit in locally advanced HNSCC using concurrent Avelumab immunotherapy with definitive standard chemo-radiotherapy, there are currently promising phase III trials such as IMvoke that propose the therapeutic sequence used for lung cancer in the PACIFIC trial [92,93,94,95].

The role of p53 is once again at the forefront with the introduction of immunotherapy in the multimodal treatment of cancers. Intracellular accumulation of hotspot mutations could be immunogenic, resulting in the triggering of p53 neo-antigens associated with T lymphocyte-mediated intra-tumoral immune responses. The potential to use p53 antigens as therapeutic targets is proposed by Chasov et al., with monoclonal antibodies (mAbs) being part of the new strategy [96].

The accumulation effect of mutant p53 in the cancer cell induced by multiple mechanisms of viral infections, such as the degradation of p53 WT and inhibition of the Rb protein, has already been demonstrated as being related to tumor apoptosis. Not associated with the characteristic regulatory mechanisms of p53 WT, including the inability to bind to DNA to promote MDM2 transcription, mutant p53 accumulated in the cell will become an active antigen in order to generate a more intense response to immunotherapy, resulting in cancer cell death. The expression of programmed cell death ligand 1 (PD-L1), one of the preferred targets of immunotherapy, is down-regulated by mutant p53 via miR-34, thus demonstrating the involvement of p53 expression in the amplitude of the response to ICI therapy [97,98,99].

Although KRAS/ATM/EGFR/STK11 co-mutations are considered independent predictive factors of the ICI response, not all types of p53 mutations appear to have the same predictive power. Although the missense and nonsense p53 alterations have not been mentioned before, the study by Sun et al. also evaluates these two types of mutations in relation to PD-L1 to anticipate the response to immunotherapy in lung adenocarcinoma [100].

## 8. p53, New Horizons for Head Neck Cancer Treatment

Cancer stem cells (CSC) and hypoxia are key factors in radiation resistance being evaluated in personalization strategies for head and neck cancers radiation therapy in order to increase the rate of tumor control. Doses escalation, but also an increasingly popular method (hypo-fractionation), is a possible strategy to overcome radioresistance, as even historical radiobiological studies mention. Hyper-fractionation, although considered a potential method, requires additional resources and can be difficult to implement. Marcu and collaborators use an experimental in silico model in order to evaluate the cell division probability, the average time of a cell cycle and the doubling time of the tumor volume for HNSCC. The values obtained (1.9%, 33 h and 52 days, respectively) for the variables mentioned above justify the authors’ hypothesis that incipient oxic and hypoxic tumors may benefit from hypo-fractionation, but tumors with oxygen levels below 6 mmHg and a percentage of 5.9% CSC pre-treatment require either systemic adjuvant treatment or dose escalation to 81.6 Gy. In the case of advanced tumors, hyper-fractionation is the authors’ choice in the concept of overcoming radioresistance.

Cisplatin and anti-epidermal growth factor receptor (EGFR) antibody Cetuximab are agents with demonstrated radiosensitizing potential, already included in the HNSCC therapeutic protocol. Radiotherapy with weekly Cisplatin (40 mg/m^2^) administered until a total dose of 70 Gy in 35 daily fractions over 7 weeks or bio-radiotherapy with Cetuximab for platinum non-eligible cases is currently the definitive, non-surgical, standard treatment. De-escalation of treatment in cases of HNSCC HPV+ and the use of induction chemotherapy followed by chemo-radiotherapy for cases with potentially unfavorable outcomes are options evaluated with possible benefits for carefully selected groups of patients with radiosensitive and chemo-sensitive tumors [86,96,97,98,99,100].

Interest in p53 as a modulator of radiosensitivity in head and neck cancers is not a recent research concern. A historical hypothesis associating the p53 mutation with the possible increase in radiosensitivity in HNSCC has been contradicted more than two decades after the study conducted by Brachman et al. Ras, myc, and raf expression mutations have been associated with radioresistance, thus demonstrating that the p53 mutation is not directly involved in tumor sensibility to irradiation [101,102].

DNA methyltransferase 3B (DNMT3B) has been shown to be a tumor genesis-associated factor in nasopharyngeal cancer, but its involvement in radioresistance is still poorly understood. The study by Wu et al. is the first to mention the ability of radiotherapy to induce DNMT3B and implicitly radioresistance and therapeutic failure. Silencing of DNMT3B can reduce migration and invasion by inhibiting EMT. Consequently, the authors propose a complex strategy that combines p53 and p21 by demethylation to generalize apoptosis and cell cycle arrest with a direct effect on increasing the radiosensitivity of nasopharyngeal cancer tumor cells [103].

PD-L1 bound to the p53 protein is thought to influence both prognosis and response to treatment, particularly ICI. The study by Tojyo and collaborators evaluates the correlation between cytokeratin 17 (CK17), PD-L1, and p53 and its value as a diagnostic and prognostic biomarker of HNSCC. Analyzing data obtained from 48 patients with squamous cell carcinoma of the oral cavity PD-L1, p53 and CK17 were evaluated regarding the possible clinical and pathological correlation. p53 status was associated with tumor stage T, TNM stage and PD-L1 expression, but CK17 was not correlated with p53 or prognosis [83].

HNSCC associated with HPV infection has different biological, clinical, and therapeutic characteristics compared to classic HNSCC, which is generally associated with a long history of smoking. At the molecular level, there are differences that may explain different prognoses, evolution, and response to therapeutic agents, whether it is chemotherapy, biological therapy, radiotherapy, or ICI therapy. The p53 gene is not usually mutant in HPV+ HNSCC, but the E6 viral oncoprotein has the effect of inhibiting and proteasomal degradation of HPV-induced p53. A striking difference between HPV-treated HNSCC subtypes is that the tumor suppressor gene TP53 is not usually mutated into HPV+ cancer cells. However, p53 is inhibited by the viral oncoprotein E6, leading to premature proteasomal degradation for this cancer subtype. Implications with potential therapeutic results in the design of methods to restore p53 function. Bortezomib, a proteasome inhibitor active in the treatment of multiple myeloma and mantle cell lymphoma, has been evaluated in cell lines in HNSCC HPV +, demonstrating its ability to restore p53 function and hypothetical restore radiosensitivity via p21/p53 transactivation. However, in combination with radiotherapy or chemotherapy with Cisplatin, Bortezomide has no radiosensitivity and chemo-sensitivity modulator effect on HNSCC HPV+ and HPV- cell lines [2,104,105,106,107,108,109,110].

For HPV- HNSCC multiple strategies are proposed for the restoration of p53 function in order to modulate radiosensitivity. Targeting factors that degrade, inhibit or prevent p53 WT breakdown, such as PM2, RITA nutlin-3, Ch1iB, MDMX/4, but also direct modulators of p53 binding and reactivation such as COTI-2, PRIMA-1, CP-31398, APR-246 are options proposed and evaluated for HNSCC HPV- [111].

C-MYC, whose positive expression was identified in 35.7% of HNSCCs, is considered a mediator of p53 GOF, being associated with radioresistance of head and neck cancers, but the association of C-MYC with p53 is also a negative prognostic factor. BYL719 (alpelisib), a PI3Kα-selective inhibitor, is proposed as a therapy for restoring radiosensitivity by breaking the C-MYC p53 bond. A chemical chaperone (glycerol) has the potential to restore p53 function on HNSCC cell lines, being considered an agent with a possible role in p53-mediated radiosensitization [107,108,109].

CIP2A is considered another possible therapeutic target of rapamycin for inducing senescence in HNSCC radioresistant cells. The presence of large amounts of CIP2A in HNSCC radioresistant cells with mutant p53 justifies strategies for modulating radioresistance by controlling this pathway [110].

Oct4 and CIP2A in combination are considered to be potentiating factors for radiation resistance in head and neck cancers, and Ventelä and collaborators propose the use of the Oct4/CIP2A combination as a biomarker for the selection of radiation-resistant tumors. Analyzing p53, NDFIP1, EGFR and nuclear positivity of stem cell Oct4 marker and CIP2A, Routila et al. did not identify a correlation between p53, EGFR, CIP2A and intrinsic radiosensitivity, but stem cell Oct4 and NDFIP1 were correlated with radioresistance in HNSCC [112].

Hyperexpressed p53 could trigger the abscopal effect, with a case of oligo-metastatic hypopharyngeal cancer with a systemic response being reported. p53 GOF could be associated with the presence of this effect in RAS mutant melanoma cases [113,114]. The study by Klinakis et al. hypothesizes that mutant TP53 is more common in primary tumors than in distant metastases, and the impact of TP53 mutation in metastatic disease regarding ICI treatment was also assessed. The study included 512 primary tumor biopsies and 134 distant metastases biopsies, all from HNSCC. The results indicate a lower frequency of TP53 mutations in metastatic disease but the predominance of missense mutations. A higher TMB in metastases than in primary tumors also justified an unfavorable response to immunotherapy for primary tumors. Ginkel’s study highlights a 95% and a 91% concordance of p53 mutation in distant metastases, respectively, in loco-regional recurrences, by analyzing a lot of 239 HNSCC. Authors propose the use of p53 as a biomarker of response to treatment. A stratification of the prognosis and prediction of the response to chemotherapy and radiotherapy is also proposed by Zhou et al. based on p53 status, with a focus on the possibilities of GOF for both WT and mutant p53 HNSCC cases [110,111,112,113,114,115,116].

MiR-125a is a modulator of cell proliferation migration and invasion via p53. Although there are 49 miRNAs that can discriminate p53 WT from mutant p53 in HNSCC, the involvement of miRNA in the radiosensitivity of these head and neck cancers is less reported [117,118,119].

Variations in cell death rates and radiosensitivity of tongue SCC classified according to p53 status after X-rays (low-linear energy transfer (LET)) or carbon-ion beams (high-LET heavy ion) irradiation were evaluated by Asakawa and collaborators two decades ago. The study highlights a significant dependence of p53-mediated radiosensitivity depending on the ionizing radiation type used. A limited ability to modulate radiosensitivity and a lower rate of apoptosis associated with X-rays were used to explain the authors’ conclusion. In the case of carbon ion therapy, the biological effect of irradiation does not involve the p53 pathway. The results have become significant in current clinical practice with increasing interest in heavy ion radiotherapy in head and neck cancers, especially in the HPV- subtype [119,120].

The link between p53 and mitophagy has been investigated in head and neck cancers to propose a concept of modulation of radioresistance. P53-null radioresistant cells are associated with increased glycolysis and impaired mitochondrial function. p53 WT radioresistant cells are associated with a slightly altered metabolic profile and the preservation of mitochondrial integrity. These findings may promote p53 status as a biomarker to anticipate response to anti-glycolytic therapies in head and neck cancers [121].

Loss of p53 function creates an Achilles heel in HNSCC, as observed by Wilkie et al. by potentiating the Warburg effect. Loss of p53 function or mutation or down-regulation of p53 causes a lack of metabolic flexibility, malignant cells being more dependent on glycolysis by losing the ability to oxidative phosphorylation. The authors propose a strategy to increase the radiosensitivity of HNSCC to HNSCC cells with impaired p53 function if a glycolysis inhibitor is used [68].

A preclinical study using human hypopharyngeal cancer xenografts in vivo demonstrated the effect of chemo-sensitivity to Cisplatin when co-expressing growth inhibitor protein 4 (ING4) and P53. Increased sensitivity to Cisplatin associated with Ad-ING4-P53 gene therapy in hypopharyngeal cancer xenografts may be associated with induction of apoptosis by up-regulation of Bax and down-regulation of Bcl-2. Overexpression of miR-182-5p is associated with radiosensitization of ROS-mediated HNSCC cells via sestrin2 (SESN2), a molecule of oxidative stress. Overexpression of miR-182-5p has been associated with the elimination of SESN2 and radiosensitization of HNSCC. miR-182-5p is therefore involved in the tumor response to radiation mediated by oxidative stress. In bladder cancer, radiosensitization caused by curcumin is increased by p53-mediated miR-1246, the substrate being mechanisms involving G0/G1-phases and apoptosis [68,119,120,121,122,123,124,125].

Without intending to cover the vast number of studies evaluating the involvement of p53 in HNSCC, we have synthesized data on all anatomical structures of the head and neck. We also tried to synthesize in Table 1 and Table 2 different implications of the p53 pathway and some suggestive studies evaluating p53 as a modulator of radiosensitivity in HNSCC [Table 1 and Table 2] [87,126,127,128,129,130,131,132,133,134,135,136,137,138,139,140,141,142,143,144,145,146,147,148,149,150,151,152].

P53 acquires new valence due to testing of agents with the potential to restore/control p53 function for potential clinical benefit. Regardless of whether genotoxic agents or inhibitors of the p53 pathways, including the Warburg effect, are used, steps forward have been made for testing strategies that target p53 in clinical practice for both HNSCC HPV- and the viral etiology HPV+ subtype. Table 3 and Table 4 include anti-p53 target agents with radiosensitization and chemo-sensitization, respectively, and clinical trial initiatives, including p53 pathway modulation therapies [Table 3 and Table 4] [153,154,155,156,157,158,159,160,161,162,163,164,165,166,167,168,169,170,171].

## 9. Conclusions

P53 is implicated in HNSCC radiosensitivity, as evidenced by studies over the past 30 years. However, new advances in molecular biology and genetics include the involvement of miRNAs in the control of the p53 pathway, the understanding of GOF mechanisms and loss of function, and the development of different methods to restore p53 function both for HVP− and HPV + HNSCC cases. The different ratio between mutant p53 in the primary tumor and distant metastasis originating HNSCC may serve to select the best therapeutic target for activating an abscopal effect by irradiation as an activator of the immune system. p53 may be a key player in choosing the radiotherapy fractionation regimen. Evaluation in clinical trials of targeted therapies that can restore/modulate p53 function opens new horizons for synergistic associations with chemo-radiotherapy and radiotherapy for sensitization. The possibility of using the status of HPV, p53, and miRNAs as biomarkers for the selection of therapy, as well as updating the interest in tumor metabolism and, in particular, the Warburg effect as a possible target involving the restoration of p53 function and implicitly the net benefit in the therapeutic response. With increasing interest in high-LET in clinical practice, the involvement of the p53 pathway in the different radiobiological responses to HNSCC, depending on the type of ionizing radiation chosen, could argue for the use of heavy ion therapy, especially for HPV + cases. CSC, hypoxia and modulation of the p53 pathway tested in silico models may underpin the future concept of precision radiotherapy in terms of dose, fractionation, radiation type and association with systemic therapies for HNSCC. 

## Figures and Tables

**Table 1 diagnostics-12-03052-t001:** Various roles of p53 in different anatomical sites of head and neck cancers.

P53 in Head and Neck Squamous Cell Carcinomas (HNSCC)			
Cancer Type	Mechanism of Action	Results/Clinical Implication	References
All types of HNSCC	36–39 TP53 mutations detection	not specified	Peltonen et al., 2010 [126]
All types of HNSCC	identification of p53 as the most common somatic mutation	biomarker for prognosis and a predictor of clinical response to radiotherapy and chemotherapy	Zhou et al., 2016 [117]
All types of HNSCC	identification of p53 TP53 mutations in DNA-binding regions (L2, L3 and LSH motif)	marker for predicting prognosis and response to radiation	Peltonen et al., 2011 [127]
All types of HNSCC	TP53 mutation detection in 53.3% of patients	TP53 mutation is associated with reduced survival	Poeta et al., 2007 [128]
HNSCC treated surgically with curative intent	HPV16-positive and p53 mutation coexistence	possible implications for patient outcomes	Westra et al., 2008 [129]
Oral cavity, oropharynx or larynx surgically treated	TP53 mutation detection	tobacco and alcohol consumption correlation, tumor histological grading correlation, no correlation with T and N stages and no clinical correlation specified	Golusinski et al., 2016 [130]
HNSCC with radical tumor resection	p53 mutations detection in surgical margins	identification of p53 in surgical margins as a prognostic factor for high recurrence risk	van Houten et al., 2002 [131]
Oral cavity SCC (OCSCC)	P53 mutation detection	not specified	Ragos et al., 2018 [132]
OCSCC	P53 identification in relation with carcinogens	high incidence of P53 mutation in tobacco users.	Lazarus et al., 1996 [133]
OCSCC	betel quid chewing, alcohol use and smoking in relation to the p53 mutation	not specified	Hsieh et al., 2001 [134]
All types of HNSCC	correlation of the 36 TP53 mutations confirmed with carcinogens	smoking, alcohol and work history and no clinical correlation specified	Peltonen et al., 2010 [126]
Nasopharyngeal carcinoma (NPC)	identification of p53 protein in NPC primary tumor and metastatic nodes	no statistically significant correlation with p53 immuno-reactivity and overall and disease-free survival was identified.	van Houten et al., 2002 [131]
All types of HNSCC	loss of p53 function	adrenergic trans-differentiation of tumor-associated sensory nerves with inhibition of tumor growth as a consequence	Amit et al., 2020 [136]
All types of HNSCC	overexpression of p53 protein was detected	not specified, only the association of the p53 mutation with carcinogens such as tobacco is mentioned	Somers et al., 1992 [137]
		p53 mutations are uncommon in virus-related HNSCC but common in oropharyngeal and hypopharyngeal carcinoma	Maruyama et al., 2014 [138]
All types of HNSCC	p53 protein degradation by the viral oncoprotein E6 and p53 mutations inHPV16-positive tumors	inverse relationship between human papillomavirus-16 infection and disruptive p53 gene mutations, clinical implications not specified	Westra et al., 2008 [129]
NPC	to determine if the mutation in p53 participates in the development of the malignant clone	p53 gene are unlikely to be involved initially contributing to clonal outgrowth, clinical implications not specified	Effert et al., 1992 [139]
All types of HNSCC	restoring the tumor suppressor activity of p53	Ad-E6/E7-As and bortezomib may restore p53 function to improve therapeutic outcomes	de Bakker et al., 2022 [2]
Larynx HNSCC	evaluation of p53 as a predictor for larynx preservation	p53 and Bcl-xL are strong predictors of larynx preservation after induction chemotherapy and radiotherapy	Kumar et al., 2008 [78]
All types of HNSCC	evaluation of differences in the mutation profile of TP53 in primary and metastatic disease	TP53 mutations are associated with higher TMB scores in only metastatic NHSCC, suggesting an unfavorable response to ICI	Klinakis et al., 2020 [116]
All types of HNSCC	evidence of concordance between p53 mutations in primary disease and metastasis	TP53 is associated with metastases, recurrence and as a post-treatment biomarker of disease evolution	van Ginkel et al., 2016 [115]
All types of HNSCC	p63 and p73 may act synergic with p53	p63 and p73 profiles modulate sensitivity to chemotherapy	Gwosdz et al., 2005 [141]
OSCC and oropharynx SCC	evaluation of Ki-67, PCNA and p53 status as prognostic factors	no relationship found between p53 or PCNA status and tumor prognosis	Sittel et al., 1999 [142]

**Table 2 diagnostics-12-03052-t002:** p53 and HNSCC—implication in radiosensitivity and radioresistance.

Cancer Type	Study Hypothesis	Radiosensitivity	Reference
All types of HNSCC	p53 tumor suppressor is associated with G1 arrest following DNA damage by X-irradiation	no effect related to p53 mutation	Brachman et al., 1993 [100]
Human squamous cell carcinoma (SCC) cell lines	comparative evaluation of radiosensitive and radioresistant cell lines which exhibited alterations of the p53 gene	ras, myc, and raf expression correlated with radioresistance, but not p53	Jung et al., 1992 [101]
All types of HNSCC	determine the incidence of p53 alterations in HNSCC refractory to radiotherapy	p53 dysfunction associated with poor response to radiotherapy and chemotherapy	Ganly et al., 2000 [143]
All types of HNSCC	proposing a new concept for mutant p53-targeting cancer therapies	glycerol-mediated restoring p53 function may increase radiosensitivity	Ohnishi et al., 2000 [108]
Human OSCC cell lines	Bax-mediated induction of apoptosis for p53 mutant cells	glycerol enhances radiosensitivity	Imai et al., 2005 [144]
OSCC	hypothesis that the p53 mutation is associated with resistance to chemotherapy and radiation therapy	DNA contact mutation of p53 could be marker radioresistance	Servomaa et al., 1996 [145]
NPC	inhibition of cell invasion and metastasis through activating the PBK-dependent p53 signaling pathway	microRNA-372 enhances radiosensitivity via the p53 pathway	Wang et al., 2019 [87]
NPC	SALL4 silencing increased radiation-induced DNA damage, apoptosis, and G2/M arrest	SALL4 induces radioresistance via the ATM/Chk2/p53 pathway	Nie et al., 2019 [146]
Epstein–Barr virus-positive NPC	LMP1 induced an increase in CSC-like CD44(+/High) radioresistant cells	cancer stem-like cells contribute to radioresistance by suppressing the p53-mediated apoptosis pathway	Yang et al., 2014 [147]
OSCC	evaluation of p53 and Ki-67 as a biomarker of response to radiochemotherapy	not confirmed as predictors of radiosensitivity and chemo-radiotherapy response	Koelbl et al., 2001 [148]
OSCC	low proliferation (Ki-67 < 20%) is	Ki-67 and p53 overexpression may predict radioresistance	Couture et al., 2002 [149]
2 HNSCC cell lines	carbon ions and x-rays induce different modes of p53-mediated cell death	high LET irradiation induced distinct types of cell death on 2 different cell lines, and different radiosensitivity may be the cause of target repopulation by modulating apoptosis.	Maalouf et al., 2009 [150]
Human tongue SCC cell lines	testing for variations in p53-dependent cell death and radiosensitivity X-rays (low-linear energy transfer (LET) or carbon-ion beams (high-LET heavy ion)	radiation-induced growth inhibition	Asakawa et al., 2002 [120]
All types of HNSCC	implication of p53 and epidermal growth factor receptor (EGFR)	radiation-mediated apoptosis by p53- and EGFR-mediated DNA repair are both factors of radioresistance	Hutchinson et al., 2020 [151]
All types of HNSCC	evaluation of 3 main pathways: EGFR, the phosphotidylinositol-3-kinase (PI3K)/Akt/mammalian target of rapamycin (mTOR), and the p53 in radiosensitivity	association of radiochemotherapy with a specific target for improving radiosensitivity	Perri et al., 2015 [152]

**Table 3 diagnostics-12-03052-t003:** p53 restoration-based therapy for radiotherapy and chemotherapy sensitivity modulation in HNSCC.

Cancer Type	p53 Function Restoration Therapy	Endpoint	References
Laryngeal wt p53 carcinoma	nutulin-3	radiosensitivity	Arya et al., 2010 [155]
HNSCC	Av1-p53	radiosensitivity	Pirollo et al., 1997 [153]
HPV+ HNSCC	Ad-E6/E7-As	sensitivity to Cisplatin	Kojima et al., 2018 [154]
Nasopharyngeal carcinoma	nutulin-3	chemosensitivity and radiosensitivity	Yee-Lin et al., 2018 [156]
HPV+ HNSCC	bortezomib	chemosensitivity- or radiosensitivity-negative results	Seltzsam et al., 2019 [104]
HNSCC OSCCC	glycerol	radiosensitivity	Ohnishi et al., 2000 [108] Imai et al., 2005 [144]
Wt p53 HPV- cells	PM2	radiosensitivity	Mortensen et al., 2019 [157]
p53 deficient cells	ONYX-015	synergy with cisplatin and 5-fluorouracil-based chemotherapy	Khuri et al., 2000 [158]
HNSCC	XI-011	increase Cisplatin sensitivity	Roh et al., 2014 [159]
All types of HNSCC	RITA	Radiosensitivity	Chuang et al., 2014 [160]
HNSCC	COTI-2	Cisplatin-based chemotherapy and radiotherapy sensitivity	Lindemann et al., 2019 [161]
HNSCC	MK-1775	Cisplatin sensitivity	Moser et al., 2014 [162] Osman et al., 2015 [163]

**Table 4 diagnostics-12-03052-t004:** Clinical trials based on p53 restoration therapy for HNSCC.

Cancer Type	p53 Function Restoration Therapy	Clinical Trial/Endpoint	References
Recurrent HNSCC	INGN 201	NCT00041626/Phase III/Cisplatin and 5-Fluorouracil sensitivity	ClinicalTrials.gov NCT00041626 [164]
HNSCC	COTI-2	Phase I/Tolerability	ClinicalTrials.gov NCT02433626 [165]
HNSCC	adenovirus-p53 gene (Gendicine) + radiotherapy	randomized controlled clinical trial/safety and efficacy	Zhang et al., 2005 [166]
Liver metastases of solid tumors and recurrent HNSCC	Ad-p53 With Capecitabine (Xeloda) or Anti-PD-1	phase 1–20/Safety and Efficacy	ClinicalTrials.gov NCT02842125 [167]
Recurrent HNSCC	Ad5CMV-p53	phase II/objective response rate	ClinicalTrials.gov NCT00003257 [168]
Newly-diagnosed stage III/IV, resectable oral cavity, oropharynx, hypopharynx, or larynx SCC	Ad5CMV-p53 gene followed by cisplatin and radiotherapy	phase II/effectiveness	ClinicalTrials.gov NCT00017173 [169]
Reccurent or metasatic HNSCC	Ad-p53 + immune checkpoint inhibitors	safety and efficacy	ClinicalTrials.gov NCT03544723 [170]
HNSCC	ONYX-015+ cisplatin/fluorouracil	Phase I/feasibility and maximum tolerated dose (MTD)—Withdrawn	ClinicalTrials.gov NCT00006106 [171]

## Data Availability

Not applicable.

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
