# Peer review of "p53 Modulates Radiosensitivity in Head and Neck Cancers—From Classic to Future Horizons"

_diagnostics, 2022, doi:10.3390/diagnostics12123052_

Round 1
Reviewer 1 Report
Good review article. Only comment is to take care of abbreviations (to elaborate when it first appear in the text) and spellings and grammar.
Author Response
Dear Reviewer,
Thank you for evaluating and correcting the manuscript and for the relevant recommendations. First of all, we removed the mentions related to the restoration of p53 function for HPV cells - which was incorrect. We have reorganized the manuscript separating the topic of chemosensitivity from radiosensitivity/radioresistance. In the chapter that refers to miRNA and the Warburg effect, I have only included references about HNSCC. We also mentioned the methodology for designing this narrative review and corrected some expressions and phrases that were expressed inappropriately/incorrectly.
In the hope that you will appreciate the proposed modifications, we are waiting for new recommendations in order to improve the manuscript.
Kind regards,
Camil Mirestean
Reviewer 2 Report
The authors write a good review on p53 as a modulator of radiosensitivity. The paper is interesting but some changes need to be made to make it publishable: - In the abstract, Line 30: expand GOF - In the introduction, the literature search strategy should be specified. It is a narrative review but the literature research performed needs to be clarified - Paragraph 2 is too long and need to be rewritten more concisely - Paragraphs 3 and 4 should be rewritten with focus on head and neck cancers and taking into account literature data based on head and neck cancers - grammar revisionAuthor Response
Dear Reviewer,
Thank you for evaluating and correcting the manuscript and for the relevant recommendations. First of all, we removed the mentions related to the restoration of p53 function for HPV cells - which was incorrect. We have reorganized the manuscript separating the topic of chemosensitivity from radiosensitivity/radioresistance. In the chapter that refers to miRNA and the Warburg effect, I have only included references about HNSCC. We also mentioned the methodology for designing this narrative review and corrected some expressions and phrases that were expressed inappropriately/incorrectly.
In the hope that you will appreciate the proposed modifications, we are waiting for new recommendations in order to improve the manuscript.
Kind regards,
Camil Mirestean
Reviewer 3 Report
It is a paper on a very relevant subject of translational research on head and neck cancer.
The English language is poor.
INTRODUCTION:
The statements in the abstract and in the introduction may lead the readers to think that HNSCC are characterized by a worse prognosis than most of the other human malignancies, it is not true, the authors could have written some more generic sentence as “besides a certain prognostic improvement in the last decades, the locoregional failure rate remains significant in head and neck cancers”.
Some statements would deserve to be better explained as the link between immune checkpoint agents and p53 (“corroborated at least with literature citations (e.g. “Recently, immune checkpoint inhibitors 51 (ICIs) entered the therapeutic spectrum of HNSCC and the potential biomarker value of 52 the TP53 gene mutation, the most common genetic mutation in these cancers, associated 53 with the accumulation of p53 in the malignant cell is investigated”). Also the statement that the strategy in HPV positive cases is restoration of p53 WT status is wrong as in most HPV positive HNSCC p53 gene is notoriously already WT.
Anyway the introduction can be read and there are some promising hints. Going on, the manuscript becomes very difficult to read and follow, the messages delivered are not clear, and no general perspective emerges. In summary one can find many sparce literature citations and disconnected sentences and concepts, but I was sorry to realize that the present work, besides the efforts, appears pointless.
Author Response

(The authors gave the same response as above.)

Round 2
Reviewer 3 Report
It is a resubmission.
Some changes have been made.
Some sentences have been moved.
Yet the manuscript remains too long, confusing, difficult to read and fail to provide a clear perspective and focus.